# The Relationship between Physical Activity and Motor Competence of Foundation Phase Children in Wales during the School Day

**DOI:** 10.3390/children11060629

**Published:** 2024-05-24

**Authors:** Amanda John, Nalda Wainwright, Jacqueline D. Goodway, Andy Williams

**Affiliations:** 1Wales Academy for Health and Physical Literacy, University of Wales Trinity Saint David, Carmarthen SA31 3EP, UK; amanda.john@uwtsd.ac.uk (A.J.); n.wainwright@uwtsd.ac.uk (N.W.); a.williams@uwtsd.ac.uk (A.W.); 2Department of Human Sciences, The Ohio State University, Columbus, OH 43210, USA

**Keywords:** physical activity (PA), motor competence (MC), fundamental motor skills (FMS), early childhood

## Abstract

Early childhood is a crucial time for children to develop their fundamental motor skills (FMS), serving as a foundation for engagement in lifelong physical activity (PA). With increasing concerns over the declining levels of PA and motor competence (MC), the aim of this study was to explore the predictors of PA in children in a play-based curriculum. A secondary purpose was to explore levels of PA and MC during the school day. The final aim was to explore whether there were sex differences. Children (*N* = 94; *M*age = 68.96 months, *SD* = 8.25) in five classes from four different schools in Wales were tested on the TGMD-2, standing long jump, and MABC-2. Levels of PA were measured using ActiGraph GT3X-BT accelerometers, and 85 children met the wear time criteria. Object control (OC) skills, standing long jump, and age significantly predicted the percentage of time spent performing sedentary behaviours *F*(10,73) = 3.026, *p* = 0.003, $R^{2}$ = 0.29 (adj $R^{2}$ = 0.20) and time spent on MVPA *F*(10,73) = 3.597, *p* < 0.001, $R^{2}$ = 0.33 (adj $R^{2}$ = 0.24). Children spent an average of 48.7% of the school day performing sedentary behaviours and 9.1% performing moderate to vigorous physical activity (MVPA) and did not achieve 60 min of MVPA. The MABC revealed that 67% were below the 15th percentile. Girls spent more time than boys performing sedentary behaviours (*p* = 0.014), and boys spent more time than girls on MVPA (*p* = 0.004). Boys outperformed girls at OC skills (*p* < 0.001), while girls outperformed boys at locomotor skills (*p* < 0.001). These findings reinforce the pivotal role teachers and parents play in providing opportunities for children to be PA. OC skills and jump were positively associated with PA, emphasising the importance of developing FMS in early childhood. There were also sex disparities for both PA and MC, along with low levels of MC, highlighting the need for investing in comprehensive programmes and initiatives that prioritise the development of FMS during early childhood.

## 1. Introduction

Physical activity (PA) plays an important role in the promotion of lifelong health and well-being of children and adolescents, encompassing both physical and mental health benefits, with the World Health Organization (WHO) recommending individuals aged 5–17 years should aim for at least 60 min of moderate- to vigorous-intensity physical activity (MVPA) daily [1]. However, despite these recommendations, globally, children, adolescents, and even adults are failing to meet these recommended levels of activity [2,3]—a trend that was already of significant concern before the COVID-19 pandemic [4]. This issue is particularly pronounced in Wales, where it has been suggested that only 51% of children aged 3–17 met the recommended PA guidelines [5]. Recent evidence suggests that PA levels have further decreased [6]. This concerning trend not only impacts physical health but also has implications for mental health, quality of life, and overall well-being among individuals of all ages [1,3]. Addressing this issue is paramount for safeguarding the health and well-being of children, adolescents, and adults alike in Wales.

The acquisition of movement patterns and physical abilities necessary for engaging in physical activities is referred to as motor competence (MC) [7]. The escalation of inactivity and sedentary behaviour likely contributes to the global decline in children’s MC levels. Despite the expectation that children should attain proficiency by the age of 7, many are falling short of reaching adequate competence levels [8]. The association between PA and MC is firmly established in the literature. Early work by Seefeldt [9] introduced the concept of a ‘proficiency barrier’, suggesting that failure to achieve a certain level of fundamental motor skill (FMS) by middle childhood could hinder the development of more advanced movements and application to sport or activity-specific movements. This inability to overcome the hypothetical proficiency barrier was believed to have repercussions on overall child health-related fitness [9]. Building upon this idea, Clark and Metcalf [10] proposed the ‘mountain of motor development’, emphasising the importance of children acquiring a broad ‘base camp’ of movement skills to enable participation in various activities. Stodden and colleagues [11] further elaborated on this notion by formulating a model that illustrates the intricate relationship between MC and lifelong PA. Their model highlights early childhood as a critical window, emphasising that opportunities for PA during this stage drive MC development and set the stage for a positive health trajectory. This model comprehensively considers factors such as perceived MC, health-related fitness (HRF), and weight status [11].

While limited evidence initially supported or refuted the model’s hypotheses proposed by Stodden and colleagues [11], subsequent studies have provided substantial support for the relationship between MC, perceived MC, and PA [12]. The model integrates perceived MC as a mediator between actual motor skill level and PA. Perceived MC reflects one’s awareness of actual movement capabilities, evolving over time with development and cognitive capacity [13,14]. During early childhood, cognitive limitations hinder accurate self-assessment of MC due to the preoperational phase of cognitive development [7]. As children progress into middle childhood and adolescence, their awareness of their physical abilities influences their engagement in physical activities, mediated by perceived competence. Individuals with high motor skills tend to choose higher levels of PA, sports, and games, viewing themselves as proficient movers and maintaining higher activity levels. Conversely, those with low MC are less inclined to participate in physical activities, often opting for more sedentary behaviours, perceiving themselves as less skilled in movement [11].

It is evident from the existing research and developmental models that early childhood is a critical period for children to develop their MC, laying the foundations for lifelong engagement in PA [10,11,12]. MC not only predicts levels of PA but is also associated with broader health outcomes such as healthy weight status [15,16] and cardio-respiratory fitness [17,18]. It is also associated with cognitive and academic outcomes [19] and strongly predicts children’s achievement of school readiness [20]. Evidence from the UK suggests a decline in PA levels among both boys and girls around the age of six to seven years as they transition into middle childhood [21]. Additionally, research suggests that boys aged 3 to 8 spend less time performing sedentary behaviours and more time in MVPA compared to girls [22,23,24]. Reviews also demonstrated a positive association between FMS and MVPA during early childhood [25,26]. This reinforces the importance of understanding and fostering MC, particularly in early childhood, for promoting lifelong PA and overall health.

The significance of movement in early childhood is strongly echoed within the Foundation Phase—a play-based curriculum for children aged three to seven in Wales. At the heart of this curriculum is the holistic development of the child, promoting a balance between adult-led instruction and child-initiated activities while utilising both indoor and outdoor environments [27]. During this age, children begin to develop their MC through FMS, encompassing locomotor (e.g., running, skipping, and jumping); object control (OC), also referred to as manipulative (e.g., throwing, kicking and catching); and stability (e.g., balancing) [7]. The Foundation Phase framework emphasises that through movement and the use of equipment, children will develop their motor skills along with balance and coordination [27]. However, research in Wales suggests that children are not developing these skills as they should, particularly OC control skills [28]. 

There is often a misconception that these skills develop naturally [29], yet studies have found that children participating in free play activities alone do not ‘naturally’ develop FMS (e.g., [30,31,32,33,34]). This is of particular importance in terms of OC skills, which involve higher perceptual demands and greater complexity in their components compared with locomotor skills. They are more complex and necessitate equipment, structured practice, instructional guidance, feedback, and reinforcement [29,35]. Globally, boys tend to exhibit higher OC skill proficiency than girls, while there tends to be no difference in locomotor skills [8,36]. These differences are likely influenced by cultural factors, such as societal expectations and available activities [37]. Addressing these gaps in skill development is crucial, as OC skills in childhood are not only associated with but predictive of participation in PA during adolescence [38,39]. Moreover, evidence suggests there is a positive association between OC skills and MVPA among preschool children [26].

According to Fisher et al. [40], environmental factors primarily account for variations in activity levels in daily life. While children have break times throughout the school day, most of their time is controlled by the teacher, who, as such, plays a key role in providing opportunities for children to be physically active. Parents also play a key role in the development of young children’s PA and FMS as they direct what they do outside of school [40,41]. Considering the play-based nature of this early childhood curriculum in Wales, it may be expected that children engage in PA throughout the day, thereby fostering the development of their MC [11]. There is, however, limited research assessing the PA and FMS levels of children in Wales during the school day. Studies conducted in other countries during the school day with children of a similar age group found sedentary behaviour ranged from 51.2% to 80% of the school day, with MVPA accounting for only 6.4% to 8.8% of the school day [22,42,43]. Given the escalating concerns surrounding declining PA levels and MC among children, the aim of this study was to explore the relationship between PA and MC. Secondly, it examined levels of MC and PA, with a specific focus on MVPA and sedentary behaviour, during the school day. Finally, it investigated if there were any sex differences between PA and MC. The following hypotheses were tested:

**Hypothesis 1.** 
*MC will significantly predict sedentary behaviour and MVPA.*


**Hypothesis 2.** 
*Boys will spend more time in MVPA than girls, while girls will spend more time performing sedentary behaviours than boys.*


**Hypothesis 3.** 
*Boys will demonstrate higher levels of OC skills than girls.*


## 2. Materials and Methods

### 2.1. Context and Participants 

The current study is situated within the larger research agenda of a three-year European-funded PhD project, led by John et al. [44], which evaluated the extent to which early childhood teachers could be trained in a programme of professional development and impact MC outcomes and the PA of young children. The current study is a sub-analysis of these data. Four schools in West Wales were recruited through convenience sampling. Two of the schools were situated in rural areas, while the other two were situated in small towns. The classes had to be situated within the Foundation Phase and include children aged between 4 and 7 years old. There were a total 5 classes in the study; 1 school had a Welsh and English language class. Thus, there was the inclusion of 2 classes from 1 of the schools. All children within the selected classes were invited to participate, resulting in an 84% response rate, with a total of 94 participants (*M*age = 68.96 months, *SD* = 8.25). Eighteen per cent of participants had an Additional Learning Need (ALN), 24% of the participants were eligible for free school meals, and 36.2% were primarily taught through the medium of Welsh. A breakdown and overview of each class are presented in Table 1.

### 2.2. Instrumentation

MC was evaluated using both process and product measures [12,24,45,46]. The process measure utilised was the Test of Gross Motor Development Second Edition (TGMD-2) [47]. The product assessment used the Movement Assessment Battery for Children—Second Edition (MABC-2) [48] (Henderson et al., 2007), and 98% (*n* = 92) of participants completed this measure. Additionally, a standing long jump for distance (measured in centimetres) standardised by standing height was included [49,50]. This is a complex skill; it requires immense leg strength, core strength, and explosive power, along with dynamic balance and multi-limb coordination for an individual to propel themselves off the ground horizontally for distance [7]. PA was evaluated by accelerometry, with only 90% of the sample (*n* = 85) meeting the inclusion criteria. Anthropometric measures were taken (*N* = 94). 

#### 2.2.1. TGMD

The TGMD-2 includes two subscales, locomotor and OC skills, totaling twelve skills. In the locomotor subscale (number in parentheses represents total possible score for the skill), the skills include run (8), gallop (8), hop (10), leap (6), horizontal jump (8), and slide (8). The OC subscale involves striking a stationary ball (10), stationary dribble (8), catch (6), kick (8), overarm throw (8), and underarm roll (8). Each skill consists of three to five critical elements or components, with a scoring system where correctly performing a component earns a score of one, while incorrect performance results in a score of zero [47]. Prior to each trial, the lead researcher provided a demonstration of the skill following the TGMD-2 protocol. Then, the child completed a practice trial, followed by two coded trials of each skill. The scores from both trials were combined to determine a score for each child in the respective subscales. These subscale scores ranged from 0 to 48 for locomotor skills and OC skills, resulting in a total motor score ranging from 0 to 96 [47]. Raw scores were utilised for analysis since the normative data for the TGMD2 was based on a U.S sample and did not correspond with the participant population in this study. Additionally, reporting raw scores offered a measure of each child’s proficiency in performing each skill relative to its critical components of proficient performance. Each trial was video-recorded for later analysis. 

Two coders with prior experience coding the TGMD, one a PE teacher and experienced coach, the other a primary school teacher and experienced coach, were trained by the lead researcher. This involved the following:Performing and practicing each skill and identifying the critical elements;Coding a member of the coding team performing the skill;Observing videos and identifying the performance criteria of each TGMD-2 skill with the expert coder and discussing why the score was awarded or not;Coders practicing on videos previously coded by an expert. The expert then provided feedback followed by a discussion;The development of ‘Gold Standard’ videos with a 97% inter-rater reliability, created by the expert and lead researcher. These videos were analysed by another expert coder and had 92% reliability. Following discussion, the ‘Gold Standard’ videos were agreed upon. The two coders completed the ‘Gold Standard’ videos and had a 95% inter-rater reliability overall.

During coding, if there were elements of the skill that were unclear, these videos were referred back to the experts for further evaluation. Thirty per cent of footage underwent random checks for reliability by the lead researcher and expert. The first coder achieved 95% agreement, and the second achieved 90% agreement. Overall inter-rater reliability for the double-blind sample was 92.5%, exceeding the required 90% threshold [51].

#### 2.2.2. Movement ABC

The Movement ABC is divided into three age bands: age band 1 (3–6 years old); age band 2 (7–10 years old), and age band 3 (11–16 years old); the age band used for this study was age band 1. The test was split into the following 3 components and involved 8 tasks (parentheses represent how they were scored):Manual Dexterity: Posting coins (time); threading beads (time); drawing trail (errors).Aim and Catch: Catching a beanbag (number of catches); throwing a beanbag onto the mat (number of hits).Balance: One-legged balance (time); walking with heels raised (number of steps); jumping on mats (number of consecutive jumps).

Following the standard protocol for the MABC-2, children were assessed individually, taking approx. 20–30 min. Prior to each trial, the lead researcher provided a demonstration of the task in accordance with the manual [48]. The child then completed a practice trial, followed by two formal trials. Inter-rater reliability was determined with one of the research team, who is an expert in physical education and motor development, with 100% agreement. They scored independently alongside the lead researcher for 20% of participants. Raw scores were converted to item standard scores for the three components subscales based on age and sex. The component score was then converted into an overall standard score (1 to 19) and percentile. For the total test score, the three component scores were added together and then converted into a standard score (1 to 19) and percentile [48]. Unlike the TGMD-2, the MABC-2 was based on a UK population; therefore, standard scores were reported for all three components and the total test score.

#### 2.2.3. Standing Long Jump

Jumping distance was measured, to the nearest centimetre, from the toe of the front foot at the starting position to the heel of the backmost foot upon landing. Prior to each trial, the lead researcher provided a demonstration of two-footed jump for distance from behind the line. Children were instructed to jump as far as they could. Following a practice trial, they then completed two trials, with the best score being recorded in cm. Distance jumped was normalised to standing height, resulting in a score that gave a percentage of standing height a child jumped.

#### 2.2.4. Physical Activity Measures

PA behaviours were measured using ActiGraph GT3X-BT accelerometers (ActiGraph Corp., Pensacola, FL, USA), which captured and recorded continuous, high-resolution PA information. Children wore the monitor on their right hip using an adjustable belt for one-week intervals, specifically from Monday to Friday (five days) during school hours (9 a.m. to 3 p.m.). The lead researcher ensured that the monitoring occurred during a standard school week with no trips, special events, or testing, maintaining a typical ‘business as usual’ environment. The class teacher and support staff were tasked with fitting the devices each morning and removing them at the end of the day. Prior to implementation, they received a demonstration along with basic guidelines on how the devices should be worn. They would check throughout the day that the monitors were in the correct position and remained on the right hip.

The ActiLife 6.13.4 software program (ActiGraph Corp., Pensacola, FL, USA) was used to initialise, download, process, and store accelerometer data. The data were recorded in 15 s epochs at 100 Hz and processed using the cut points developed by Evenson et al. [52]. The epochs were categorised into sedentary, light, moderate, vigorous, and total MVPA. This paper will report the results of sedentary behaviour and MVPA. It was decided to focus on MVPA due to the recommendation of 60 min of MVPA a day by WHO and the important role it plays in the promotion of lifelong health and well-being of children [1]. We were also interested in sedentary behaviour due to the concerns in Wales regarding decreasing levels of PA in children [5,6]. The wear time validation was conducted using the Choi et al. [53] algorithm, with data then converted from minutes into a percentage of time spent performing sedentary behaviours or MVPA of the school day in order to account for differences in school day time. The criteria for valid accelerometer data required a minimum of three days during the school week, each day having a minimum wear time of five hours (e.g., [22,43]). Data failing to meet these criteria were excluded from the final analyses, resulting in the removal of a total of 9 cases with a total of 85 participants’ data utilised in the analyses.

#### 2.2.5. Anthropometric Measures

Anthropometric measures were carried out by the lead researcher and School Nurse Support Workers who were already assigned to the schools. Participants were instructed to remove their shoes and any heavy clothing before undergoing height and weight measurements. They were positioned with their back to the stadiometer with feet together and were directed to look straight ahead so the head was in a neutral position (coronal plane). Height was measured to the resolution of the height rule (i.e., nearest millimetre/half a centimetre). For weight measurement, participants were asked to stand at the centre of the platform to ensure even weight distribution. Weight was recorded to the nearest 0.1 kg or 0.2 kg using a scale. The height and weight data were used to calculate the body mass index (BMI), with thinness, overweight, and obesity classifications based on the International Obesity Task Force (IOTF) BMI cut-offs proposed by Cole and Lobstein [54].

### 2.3. Procedures

A full ethics proposal was accepted by the University of Wales Trinity St David (UWTSD). The University Code of Ethics and the British Educational Research Association (BERA) Ethical Guidelines for Educational Research [55] were adhered to throughout the study. All data were collected during the Autumn term between September and October. Prior to this, consent was sought first from the head teachers and custodial caregiver of each child participating, and verbal assent was obtained from each child during the MC tests and before wearing the belts. 

### 2.4. Data Analysis

All data analyses were conducted using SPSS 26.0 package, and statistical significance was set at *p* < 0.05. Prior to carrying out the statistical analyses, a cleaning process was conducted to check for any errors and normality (via the Kolmogorov–Smirnov test). Descriptive analyses were conducted for all variables. 

The dependent variables for TGMD-2 (FMS GMQ, locomotor, and OC skills) and MABC-2 (total MABC, manual dexterity, aim and catch, and balance) were not moderately correlated and did not meet the assumptions to run a multivariate analysis of covariance (MANCOVAs); thus, separate analyses of covariance (ANCOVA) were conducted for MC variables to determine whether there were sex differences. Age was accounted for as a covariate (when raw scores were used) due to the age range of the children. Class was also a covariate due to the children being nested within their class. A crosstabs analysis was undertaken for MABC-2 percentile cut-off points. 

The PA variables were correlated; thus, multivariate analysis of covariance (MANCOVA) was conducted for MVPA and sedentary behaviour to determine whether there were sex differences. 

Multiple regressions were conducted to explore predictors of sedentary behaviour and MVPA. The first regression looked at the predictors (locomotor raw score; OC raw score; manual dexterity standard score; aim and catch standard score; balance standard score; and standing long jump standardised by height, age, and BMI) of per cent of the school day spent performing sedentary behaviours. Using the same predictors, the second regression focussed on predicting per cent of school day spent in MVPA.

## 3. Results

### 3.1. Levels of Physical Activity and Motor Competence 

The mean percentages of the school day spent performing sedentary behaviours and MVPA are presented in Table 2.

Descriptive statistics for age, sex, TGMD-2, standing jump, and BMI are presented in Table 3.

A crosstabs analysis was undertaken with MABC-2 percentile cut-off points to examine the extent to which children demonstrated a delay in MC. These were categorised into the following:Green: above the 15th percentile and classed as having no movement difficulty;Amber: 6th to 15th percentile and at risk of having a moving difficulty;Red: 5th percentile or below and classed as having a significant movement difficulty.

The results in Table 4 indicate that only 33% of children were above the 15th percentile. The remaining 67% were classed as at risk or having a significant movement difficulty.

### 3.2. Sex Differences in Physical Activity and Motor Competence

The MANCOVA for PA revealed significant sex differences (*F*(2,78) = 4.69, *p* = 0.012, $\eta^{2}$ = 0.11). Univariate follow-up analyses revealed a significant difference between sexes for both sedentary behaviours (*F*(1,79) = 6.25, *p* = 0.014, $\eta^{2}$ = 0.07) and for MVPA (*F*(1,79) = 8.99, *p* = 0.004, $\eta^{2}$ = 0.10). Girls spent more time performing sedentary behaviours compared to the boys, while the boys spent more time in MVPA compared to the girls. The covariates of age (*p* = 0.065) and class (*p* = 0.581) were not significant. 

Three separate ANCOVAs comparing sex differences for TGMD variables—GMQ (Gross Motor Quotient), locomotor raw scores, and OC raw scores—were conducted with age and class as covariates. The ANCOVA analysis revealed no sex differences for overall GMQ (*p* = 0.166). There were sex differences for TGMD-2 locomotor (*F*(1,88) = 16.20, *p* < 0.001, $\eta^{2}$ = 0.16) and OC skills (*F*(1,88) = 12.93, *p* = 0.001, $\eta^{2}$ = 0.13). As illustrated in Table 3, girls outperformed boys in locomotor skills, while boys demonstrated higher proficiency in OC skills compared to girls. The covariate of class was not significantly associated with GMQ scores (*p* = 0.628) or locomotor (*p* = 0.120) or OC skills (*p* = 0.170). The covariate of age was significantly associated with locomotor (*F*(1,88) = 18.97, *p* < 0.001, $\eta^{2}$ = 0.18) and OC skills (*F*(1,88) = 17.61, *p* < 0.001, $\eta^{2}\;$ = 0.17), which was expected due to the age range of the participants.

There were no sex differences in the standing long jump (*p* = 0.576). The covariate of age (*F*(1,88) = 13.20, *p* < 0.001, $\eta^{2}$ = 0.13) was significantly associated with the jump, but class (*p* = 0.811) was not.

Four separate ANCOVAs revealed that there were sex differences (*F*(1,87) = 8.75, *p* = 0.004, $\eta^{2}$ = 0.09) in total M-ABC standard scores. The mean scores in Table 5 indicate that the girls outperformed the boys. There were also significant differences in manual dexterity (*F*(1,87) = 13.08, *p* = 0.001, $\eta^{2}$ = 0.13) and balance (*F*(1,87) = 9.99, *p* = 0.002, $\eta^{2}$ = 0.10), where, again, the girls outperformed the boys (see Table 5). There were no sex differences in the subscale of aim and catch (*p* = 0.390). The covariate of class was non-significant for all MABC variables.

### 3.3. Predictors of School-Day Physical Activity

The first regression explored whether the MC variables (see Table 6), BMI, and age predicted the percentage of time spent performing sedentary behaviours. Regression analysis revealed that the model significantly predicted the percentage of time spent performing sedentary behaviours *F*(10,73) = 3.026, *p* = 0.003, $R^{2}$ = 0.29 (adj $R^{2}$ = 0.20). OC skills, standing jump, and age were significant predictors of the amount of time children spent in sedentary behaviours during the school day.

The second multiple regression explored whether the MC variables (see Table 7), BMI, and age predicted the percentage of time spent in MVPA. Regression analysis revealed that the model significantly predicted the percentage of time spent on MVPA *F*(10,73) = 3.597, *p* < 0.001, $R^{2}$ = 0.33 (adj $R^{2}$ = 0.24). In terms of individual relationships, again, OC skills, standing jump, and age were significant predictors of the amount of time children spent in MVPA during the school day.

## 4. Discussion

### 4.1. Levels of Physical Activity and Motor Competence

Our results suggest that, on average, Foundation Phase children engaged in sedentary behaviour for 49% (176 min) of the school day (from 9 am to 3 pm), and 9% (32 min) of the day was spent on MVPA. The play-based nature of the Foundation Phase could account for why the children spent slightly less time performing sedentary behaviours in comparison with children of similar age groups in the United States [22], Indonesia [42], and Scotland [43]. Nevertheless, they still spent nearly half the day performing sedentary behaviours.

Consequently, the duration of the school day alone falls short of meeting the recommended guidelines set by the World Health Organisation and Chief Medical Officers (CMOs) for Wales, as well as the broader UK, which advocates for 60 min of MVPA daily for children and young people aged 5–18. Children would need to spend an additional 8% or approximately 28 min of MVPA to meet these guidelines. Despite opportunities to be active during break times, teachers play a pivotal role in facilitating opportunities to be active throughout the school day [40]. Additionally, it is worth noting that while some children may have access to PA opportunities before or after school, for others, the school day may serve as their primary environment for being active, especially those from low-income families. The influence of parents on children’s activities outside of school cannot be overlooked; they play a crucial role in shaping their children’s habits and behaviours related to PA [41]. Thus, recognising the combined impact of teachers and parents is essential for promoting a physically active lifestyle among children both within and beyond the school setting. Recognising these dynamics is critical in devising effective strategies to promote PA among children, especially within the school setting, where a significant portion of their day is spent. These findings highlight the importance of implementing comprehensive strategies to enhance PA opportunities within school environments to ensure that children meet the recommended activity levels and foster healthy habits for life.

More concerningly, the analysis from MABC revealed that 67% of children were at risk or exhibited a movement difficulty. Only 33% of the children were above the 15th percentile. The TGMD-2 results were also low, particularly OC skills, which supports the notion that children are falling short of reaching adequate competence levels [8]. If children are not developing their motor skills at this age, they are unlikely to participate in PA in later life [11]. These findings are concerning, particularly within the context of the holistic development emphasised in this early childhood curriculum in Wales. MC not only correlates with health outcomes such as healthy weight status [15,16] and cardio-respiratory fitness [17,18] but also with cognitive and academic achievements [19] and is a strong predictor of school readiness [20].

### 4.2. Sex Differences in Physical Activity and Motor Competence

In line with our hypotheses, the results of this study reveal significant sex differences for PA and MC. Consistent with previous literature [22,23,24], girls spent more time performing sedentary behaviours compared to boys, while boys engaged in more MVPA compared to girls. These disparities may stem from various factors, including societal norms, cultural expectations, access to sports and recreational facilities, and parental attitudes towards PA [37]. Sex stereotypes and perceptions of PA as being more suitable for boys than girls may contribute to girls’ tendency towards sedentary behaviour and boys’ inclination towards more vigorous physical activities. Understanding these complex interactions between sex and environmental influences is crucial for developing targeted interventions to promote PA among children.

Our findings indicate no significant sex differences for overall FMS GMQ (TGMD); however, differences were observed in OC skills, consistent with previous research [8,36]. Interestingly, girls outperformed boys in locomotor skills (TGMD), overall MABC score, and the subscales of manual dexterity and balance. It is important to note that there is no physiological basis for boys to outperform girls, or vice versa, during early childhood, as they are physiologically very similar [37]. Thus, the observed differences likely stem from societal and environmental factors, which teachers and parents play a significant role in influencing [40,41]. Addressing these barriers is crucial, particularly during early childhood, to ensure that both girls and boys are afforded equal opportunities to develop a broad base of motor skills and lay the foundation for a positive health trajectory [10,11].

### 4.3. Predictors of School-Day Physical Activity

Contrary to the hypothesis, not all MC variables predicted sedentary behaviour and MVPA. The two MC variables that were in line with hypothesis were OC skills and standing long jump. Children with a lower proficiency in OC skills and standing long jump exhibited higher levels of sedentary behaviour. Meanwhile, children who had higher proficiency in OC skills and jumped further spent more time in MVPA.

These findings reinforce the importance of this relationship between OC and PA and support the existing literature [26]. Addressing these gaps in skill development is crucial, as OC skills in early childhood are not only associated with but predictive of participation in PA during adolescence [38,39]. OC skills encompass abilities such as throwing, catching, and kicking, which are essential for participation in a wide range of physical activities, including team sports and recreational games. Proficiency in these skills is likely to lead to greater movement efficiency and enjoyment during physical activities, thus reducing sedentary behaviour. Children who demonstrate better OC skills are likely to have greater movement proficiency. They can then apply this proficiency to more advanced activity-specific movements or sports, thus breaking through the proficiency barrier [9] and embarking on a positive health trajectory [11].

OC skills enable individuals to engage in activities that require coordination, accuracy, and spatial awareness, thereby contributing to increased levels of PA. Locomotor skills will also be required as part of this process; however, OC skills involve higher perceptual demands and are more complex. This may explain why the process measure of locomotor skills was not a predictor of PA, but OC was. The product measure of jump, however, was a predictor of sedentary behaviour and MVPA. This reinforces the importance of utilising both product and process measures when measuring MC [12,24,45,46].

Standing long jump requires immense leg strength, core strength, and explosive power, along with dynamic balance and multi-limb coordination. These are important prerequisites to becoming a proficient mover. If children do not possess these prerequisites, then it is likely to lead to higher levels of sedentary behaviour and a negative health trajectory [11]. These findings highlight the need for targeted interventions during early childhood for children to develop their FMS, particularly OC skills, so they become more proficient movers, which may contribute towards children spending more time on MVPA and less time performing sedentary behaviours.

Age was another predictor of sedentary behaviour and MVPA. The older children demonstrated higher levels of sedentary behaviour, while the younger children engaged in more MVPA. These findings align with previous UK research, which has noted a decline in PA levels in both sexes by age 6–7 years [21]. One potential explanation highlighted in the larger study by John et al. [44] was the increased pressure teachers felt with the older children in terms of testing and having to complete work in books, which meant a shift from a play-based learning approach to a more formal and structured form of teaching. Additionally, as children reach the end of the Foundation Phase (age 7 years) and enter middle childhood, which was the case for some of the children in this study, they become more aware of their actual movement capabilities [13,14]. Those demonstrating low levels of motor competence perceive themselves as a less skilled mover; thus, they are less inclined to participate in PA, often opting for more sedentary behaviours [11,12]. With 67% of children at risk or exhibiting a movement difficulty, this potentially could be another reason why the older children spent more time performing sedentary behaviours.

### 4.4. Limitations

The sample size and use of convenience sampling was a limitation. Future research needs to consider a more randomised design with more participants covering different regions of the country. Another limitation was that the study only tracked the PA during the school day and for one week during the start of the school year. Future research should examine behaviours throughout the year and consider tracking PA outside of the school day to provide more insight.

## 5. Conclusions

In conclusion, our study highlights the intricate relationship between MC and PA among children in Wales. Despite the play-based approach of this early childhood curriculum, the school day alone falls short of meeting the recommended 60 min of daily PA. The low levels of MC are also a concern, along with the age and sex disparities, which highlights the need for early targeted intervention that may minimise these disparities and health-related gaps.

Our findings support the importance of developing OC skills in early childhood and how they significantly predict both sedentary behaviour and MVPA. Jump performance was also a significant predictor of these variables. This warrants further exploration due to its simplicity and minimal equipment requirements, offering a promising avenue for future research and serving as a potential assessment tool. Further research is warranted to examine whether early intervention for motor competence serves as an underlying mechanism driving physical activity behaviours, supplemented by longitudinal data.

Overall, this study reinforces the importance of promoting PA and motor skill development in early childhood, addressing sex disparities, and providing inclusive opportunities for PA and MC within school and community settings to support children’s holistic development and well-being. Teacher training programs should include modules on promoting motor skill development and creating inclusive physical education environments that cater to the diverse needs and abilities of all children. Parents also play a crucial role in fostering motor skill development at home by providing opportunities for active play and exploration. By encouraging their children to engage in activities that challenge their motor skills, parents can help reinforce the skills learned in school and support their children’s physical development. By offering children opportunities to practice and refine these skills in supportive environments, educators and parents can lay the ‘base camp’ for a lifetime of physical activity and health [10,11].

From a policy perspective, our findings highlight the importance of investing in comprehensive programmes and initiatives within education and the community that prioritise the development of MC in children for a lifetime of physical activity and health.

## Figures and Tables

**Table 1 children-11-00629-t001:** Breakdown of classes.

Breakdown	Class 1.1	Class 2.1	Class 2.2	Class 3.1	Class 4.1
School	1.0	2.0	2.0	3.0	4.0
Class size	19	17	27	20	29
Consents	18	17	17	17	25
Class year group	Reception, Years 1 and 2	Years 1 and 2	Reception, Year 1	Year 1	Years 1 and 2
Age Range	4 to 7	5 to 7	4 to 6	5 and 6	5 to 7
Language	English	Welsh	English	Welsh	English

**Table 2 children-11-00629-t002:** School day sedentary behaviour and MVPA.

Measure	Girls (*n* = 44)	Boys (*n* = 41)	Total (*N* = 85)
	Mean (*SD*)	Mean (*SD*)	Mean (*SD*)
% of School Day Sedentary	50.39 (6.54)	46.91 (6.66)	48.71 (6.79)
% School Day MVPA	8.34 (2.15)	9.83 (3.18)	9.06 (2.78)

**Table 3 children-11-00629-t003:** Descriptive statistics for age, sex, TGMD-2, jump, and BMI measures.

Measure	Girls (*n* = 47)	Boys (*n* = 47)	Total (*N* = 94)
	Mean (*SD*)	Mean (*SD*)	Mean (*SD*)
Age (months)	69.19 (8.65)	68.72 (7.91)	68.96 (8.25)
Height (cm)	114.80 (6.35)	115.01 (6.35)	114.93 (6.33)
Weight (Kg)	22.13 (3.71)	21.40 (3.25)	21.765 (3.49)
BMI (Kg.m^2^)	16.73 (1.78)	16.09 (1.37)	16.41 (1.61)
IOTF BMI	20.76 (4.32)	19.48 (2.92)	20.12 (3.72)
Overweight or obese (IOTF)	25.5%	12.8%	19.2%
TGMD GMQ (46–160)	64.45 (8.55)	61.57 (7.45)	63.01 (8.11)
TGMD LM raw (0–48)	20.89 (4.76)	16.89 (5.34)	18.89 (5.42)
TGMD OC raw (0–48)	9.55 (4.59)	13.02 (6.07)	11.29 (5.63)
Jump (cm)	72.31 (17.79)	72.30 (18.97)	72.31 (18.27)

Notes: BMI: Body Mass Index Score; IOTF BMI: International Obesity Task Force Body Mass Index Score [54]; GMQ: Gross Motor Quotient [47]; LM: locomotor [47]; OC: object control [47]; jump standardised by height.

**Table 4 children-11-00629-t004:** Percentiles by sex for MABC-2.

Category	Girls (*n* = 46)	Boys (*n* = 46)	Total (*N* = 92)
Green	39.1%	26.1%	32.6%
Amber	23.9%	13.0%	18.5%
Red	37.0%	60.9%	48.9%

**Table 5 children-11-00629-t005:** MABC standard scores.

Measure	Girls (*n* = 46)	Boys (*n* = 46)	Total (*N* = 92)
	Mean (*SD*)	Mean (*SD*)	Mean (*SD*)
MABC-2SS* (0–19)	6.52 (2.52)	5.26 (2.84)	5.89 (2.74)
MDSS* (0–19)	5.70 (2.31)	4.28 (2.33)	4.99 (2.41)
ACSS* (0–19)	7.09 (2.72)	7.93 (3.43)	7.51 (3.11)
BalSS* (0–19)	9.57 (4.15)	7.41 (3.16)	8.49 (3.82)

Notes: MABC-2SS*: Movement ABC (Second Edition) standard scores; MDSS*: Manual dexterity standard score; ACSS*: Aim and catch standard score; BalSS*: Balance standard score [48].

**Table 6 children-11-00629-t006:** Multiple regression results for sedentary behaviour.

Measure	*B*	*SE B*	*β*	*t*	*p*
LMRS	0.080	0.163	0.065	0.494	0.623
OCRS	−0.348	0.170	−0.273	−2.043	0.045
MDSS	0.317	0.348	0.110	0.913	0.364
ACSS	0.518	0.296	0.233	1.749	0.084
BalSS	0.137	0.244	0.077	0.563	0.575
Jump	−0.131	0.052	−0.353	−2.497	0.015
BMI	0.112	0.198	0.059	0.565	0.574
Age	0.278	0.099	0.329	2.800	0.007

Note: *B*: unstandardised beta; *SE B*: standard error for the unstandardised beta; *β*: standardised beta; *t*: *t* test statistic; *p*: probability value; LMRS: Locomotor raw score [47]); OCRS: Object control raw score [47]; MDSS: Manual dexterity standard score [48]; ACSS: Aim and catch standard score [48]; BalSS: Balance standard score [48]; BMI: International Obesity Task Force Body Mass Index Score [54].

**Table 7 children-11-00629-t007:** Multiple regression results for MVPA.

Measure	*B*	*SE B*	*β*	*t*	*p*
LMRS	0.044	0.065	0.087	0.679	0.499
OCRS	0.174	0.068	0.333	2.554	0.013
MDSS	0.012	0.139	0.010	0.083	0.934
ACSS	−0.113	0.118	−0.123	−0.954	0.343
BalSS	−0.103	0.097	−0.141	−1.056	0.294
Jump	0.072	0.021	0.475	3.452	0.001
BMI	0.005	0.079	0.006	0.059	0.953
Age	−0.086	0.040	−0.250	−2.181	0.032

Notes: *B*: unstandardised beta; *SE B*: standard error for the unstandardised beta; *β*: standardised beta; *t*: *t* test statistic; *p*: probability value; LMRS: Locomotor raw score [47]; OCRS: Object control raw score [47]); MDSS: Manual dexterity standard score [48]; ACSS: aim and catch standard score [48]; BalSS: balance standard score (Henderson et al., 2007 [48]); BMI: International Obesity Task Force Body Mass Index Score [54].

## Data Availability

The data presented in this study are available on request from the corresponding author. The data are not publicly available to protect the privacy of the participants.

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
