# Peer review of "The Relationship between Physical Activity and Motor Competence of Foundation Phase Children in Wales during the School Day"

_children, 2024, doi:10.3390/children11060629_

Round 1

Reviewer 1 Report

Comments and Suggestions for Authors

The authors have done a great job explaining the introduction, methods, presenting the data, and discussion. My only suggestion is that 1) under discussion 3.3 section Predictors of School-day Physical activity, it would be a stronger read if authors explained the significance of the predictors (eg., OC skills and long jump) for PA and sedentary behavior, what might be some mechanism, potential explanations why those predictors but not the others the outcomes, instead of implications -Teacher training programs and the role of parents. The other suggestion is that to include the limitation of the study.

Author Response

Thank you so much for reviewing our paper and providing such valuable feedback.

Please see the updates in 'red' we've taken on board your comments for the discussion and have re-worked 4.3. Your comments were really helpful and have hopefully improved this section.

We have also included a limitations 4.4.

Thank you again, if you have any further comments or feel there needs to be additional amendments then please let us know.

Reviewer 2 Report

Comments and Suggestions for Authors

Thank you for the opportunity to review the article with the title - The relationship between physical activity and motor competence of Foundation Phase children in Wales during the school day.

The article addresses a topical issue and is well structured.

Recommendations:

1. Introduction - to be added at the end of this section are the aims and hypothesis of the study.

2.1. Context and participants - the inclusion criteria of the subjects in the study should be mentioned.

4. Discussion - we recommend adding the limits of the study and highlighting the theoretical and practical implications based on the relevant results of this study.

5. Conclusions - to add future research directions in correlation with the topic and the relevant results of the study.

Author Response

Thank you so much for reviewing our paper and providing such valuable feedback.

1. Introduction - to be added at the end of this section are the aims and hypothesis of the study- Hopefully the aims are more clearer now and we've also included hypotheses.

2.1. Context and participants - the inclusion criteria of the subjects in the study should be mentioned. This has now been update and hopefully more clearer

4. Discussion - we recommend adding the limits of the study and highlighting the theoretical and practical implications based on the relevant results of this study. limitations has been added at the end of the discussion 4.4 and there are updates within the discussion. 

5. Conclusions - to add future research directions in correlation with the topic and the relevant results of the study. The conclusion has been updated to hopefully address this.

Thank you again for reviewing our paper, if you have any further comments or feel there needs to be additional amendments then please let us know.